# Satellite quantification of oil and natural gas methane emissions in the US and Canada including contributions from individual basins

Lu Shen[1,2,3*], Ritesh Gautam[3], Mark Omara[3], Daniel Zavala-Araiza[3,4], Joannes D. Maasakkers[5], Tia R. Scarpelli[2], Alba Lorente[5], David Lyon[3], Jianxiong Sheng[6], Daniel J. Varon[2], Hannah Nesser[2], Zhen Qu[2], Xiao Lu[2,7], Melissa P. Sulprizio[2], Steven P. Hamburg[3], Daniel J. Jacob[2]

[1]Department of Atmospheric and Oceanic Sciences, School of Physics, Peking University, Beijing 100871, China
[2]School of Engineering and Applied Sciences, Harvard University, Cambridge, Massachusetts 02138, United States
[3]Environmental Defense Fund, Washington DC 20009, United States
[4]Institute for Marine and Atmospheric Research Utrecht, Utrecht University, 3584 CC, Utrecht, The Netherlands
[5]SRON Netherlands Institute for Space Research, Leiden, the Netherlands
[6]Center for Global Change Science, Massachusetts Institute of Technology, Cambridge, Massachusetts 02139, United States
[7]School of Atmospheric Sciences, Sun Yat-sen University, Zhuhai, Guangdong, China 519082

*Correspondence to*: Lu Shen (lshen@pku.edu.cn)

**Abstract.** We use satellite methane observations from the Tropospheric Monitoring Instrument (TROPOMI), for May 2018 to February 2020, to quantify methane emissions from individual oil and natural gas (O/G) basins in the US and Canada using a high-resolution ($\sim 25$ km) atmospheric inverse analysis. Our satellite-derived emission estimates show good consistency with in-situ field measurements ($R=0.96$) in 14 O/G basins distributed across the US and Canada. Aggregating our results to the national scale, we obtain O/G-related methane emission estimates of 12.6±2.1 Tg a[-1] for the US and 2.2±0.6 Tg a[-1] for Canada, respectively 80% and 40% higher than the national inventories reported to the United Nations. About 70% of the discrepancy in the EPA inventory can be attributed to five O/G basins: the Permian, Haynesville, Anadarko, Eagle Ford, and Barnett, which in total account for 40% of US emissions. We show more generally that our TROPOMI inversion framework can quantify methane emissions exceeding 0.2-0.5 Tg a[-1] from individual O/G basins, thus providing an effective tool for monitoring methane emissions from large O/G basins globally.

## 1 Introduction

Increasing atmospheric methane has driven a 0.5°C global warming since 1850, making methane abatement a critical means to limit future warming (IPCC, 2021). Methane emissions have a warming potential 80 times higher than carbon dioxide over a 20-year horizon (Myhre et al., 2013; Ocko et al., 2021). Methane is the primary component of natural gas which is an increasingly important energy source in the US and Canada, accounting for a third of national energy consumption in 2019 (US International Energy Agency, https://www.iea.org/). The production of oil and gas (O/G) in the US has more than doubled since 2005 (Enverus DrillingInfo, 2020), raising concerns about the climate impacts from methane emissions. National greenhouse gas emission inventory data reported to the United Nations Framework Convention on Climate Change (UNFCCC)

by the US Environmental Protection Agency (EPA) and Environmental and Climate Change Canada (ECCC) governmental agencies report methane emissions from O/G sectors of 7.0 Tg a$^{-1}$ in the US and 1.6 Tg a$^{-1}$ in Canada in 2018 (EPA, 2020; ECCC, 2020), accounting for 13% of global O/G methane emissions (Scarpelli et al., 2020).

Emission inventories reported to the UNFCCC are based on 'bottom-up' estimates by applying emission factors to activity data. Many 'top-down' studies using measurements of atmospheric methane have shown that the national O/G methane

emission inventories in the US and Canada are biased low (Brandt et al., 2014; Alvarez et al., 2018; Omara et al., 2018; Maasakkers et al., 2021; Johnson et al., 2017; MacKay et al., 2021). Alvarez et al. (2018) estimated the US O/G methane emissions in 2015 as $13 \pm 2$ Tg a$^{-1}$ by extrapolating field observations from 9 O/G production basins, and found the emissions to be 60% higher than EPA estimates. Maasakkers et al. (2021) and Lu et al. (2022) inferred a factor of 2 underestimate in EPA oil emissions by inversion of Greenhouse Gases Observing Satellite (GOSAT) data. Ground-based and satellite

observations for the Permian Basin, the largest oil-producing basin in the US, indicate an emission source of 2.7-3.2 Tg a$^{-1}$ (Zhang et al., 2020; Schneising et al., 2020; Robertson et al., 2020; Lyon et al., 2021), 3 times higher than expected based on the EPA reported data. Field and GOSAT measurements over Canada similarly show a factor of 1.5 or greater underestimate of O/G emissions in the ECCC inventory(Johnson et al., 2017; Baray et al., 2018, 2021; Atherton et al., 2017; Lu et al., 2022). A likely reason to explain the large discrepancy between the national emission inventories and atmospheric measurements is

that the inventories do not properly account for the heavy-tailed emissions due to abnormal operating conditions and malfunctions, including fugitive emissions from venting, leakage, inefficient flaring, and blowouts (Brandt et al., 2014; Zavala-Araiza et al., 2021; Alvarez et al., 2018; Pandey et al., 2019; Duren et al., 2019; Lyon et al., 2021; Rutherford et al., 2021).

While inversion of GOSAT satellite observations has emerged as a powerful tool to quantify methane emissions from different sectors (such as O/G) on global and continental scales (Cressot et al., 2014; Alexe et al., 2015; Maasakkers et al., 2019; Lu et

al., 2021; Turner et al., 2015; Zhang et al., 2021), GOSAT data has limited skill on regional scales because individual sampling tracks are separated by ~270 km. On the other hand, field campaigns can characterize emissions on regional scales(Alvarez et al., 2018) and from point sources (Frankenberg et al., 2016; Duren et al., 2019), but they are limited in their spatial extent and temporal duration, which is problematic because of the temporal variability and intermittency of emissions (Cusworth et al., 2021; Varon et al., 2021; Lyon et al., 2021).

The Tropospheric Monitoring Instrument (TROPOMI) onboard Sentinel-5P is a more recent satellite mission that provides daily continuous methane observations starting in May 2018 (Lorente et al., 2021), considerably increasing the potential for monitoring regional methane emissions from space (Schneising et al., 2020; Zhang et al., 2020; Shen et al., 2021). Here we exploit TROPOMI to better quantify methane emissions from all major O/G production basins in the US and Canada with a high-resolution (~25 km) inversion of 22 months of data, and using the most recent gridded versions of the EPA and ECCC

inventories as prior estimates (Maasakkers et al., 2016, with updates; Scarpelli et al., 2022). Our inversion uses an analytical

method that provides closed-form error characterization as part of the solution, and enables an ensemble approach to assess the sensitivity of the results to the choices of inversion parameters and data filters. This allows us to evaluate the ability of TROPOMI to quantify methane emissions from an individual O/G basin as a function of source and observation characteristics. From there we draw general conclusions about the emerging role of satellite observations in quantifying regional methane emissions and evaluating bottom-up emission inventories.

## 2 Data and Methods

### 2.1 Satellite observations

We use the TROPOMI methane retrieval version 2.02 from (Lorente et al., 2021) for the period May 2018 - February 2020. TROPOMI is onboard the polar sun-synchronous Sentinel-5 Precursor satellite with a ~13:30 local overpass time, and provides daily global coverage in cloud-free conditions with 7 km × 7 km spatial resolution at nadir (7 km × 5.5 km since August 2019) (Hu et al., 2016; Veefkind et al., 2012). The column-averaged methane dry mixing ratio ($XCH_4$) is retrieved using the sunlight backscattered by the Earth's surface in the shortwave infrared (SWIR) 2.3 µm spectral band using the RemoTeC full-physics algorithm with near-uniform sensitivity down to the surface. We do not consider observations after February 2020 because the Covid-19 pandemic could significantly affect O/G related emissions(Lyon et al., 2021). We only use good quality $XCH_4$ retrievals that meet the following recommended criteria: (1) qa_value $\geq$ 0.5, (2) blended albedo $\leq$0.85, and (3) surface altitudes $\leq$ 2 km (Fig. S1). The blended albedo is a weighted difference of near-infrared (NIR) and SWIR albedos to filter scenes covered by snow(Wunch et al., 2011). The resulting total number of TROPOMI observations is $7\times10^6$ in the US and Canada for May 2018 – February 2020. The number of observations per 0.25° × 0.3125° inversion grid cell is typically in the 100-1000 range and exceeds 1000 in the Southwest US (Fig. S2).

When validated with the ground-based measurements from the Total Column Carbon Observing Network (TCCON), TROPOMI $XCH_4$ has a mean bias of -3.4 ± 5.6 ppb globally and 6.4 ± 4.1 ppb for the US(Lorente et al., 2021). We also intercompared the TROPOMI data with Greenhouse gases Observing SATellite (GOSAT) data from the University of Leicester version 9.0 Proxy $XCH_4$ retrieval(Butz et al., 2011; Parker et al., 2020) and find the mean TROPOMI-GOSAT difference to be -5.4±6.9 ppb in North America (Fig. S3). The mean bias in the TROPOMI data is effectively corrected in the specification of boundary conditions. The regional bias (standard deviation of the mean bias) is below the 10 ppb threshold recommended by Buchwitz et al. (2015) for successful inversions.

### 2.2 Gridded national bottom-up inventories

Prior anthropogenic methane emissions in the US and Canada are from the sector-resolved national inventories produced by the EPA (Inventory of U.S. Greenhouse Gas Emissions and Sinks)  and ECCC and spatially allocated to a 0.1°×0.1° grid by Maasakkers et al. (2016) for the US in 2012 and Scarpelli et al. (2021) for Canada in 2018. We extrapolated the US emissions

for the O/G production sector to 2018 based on upstream well data in the Enverus DrillingInfo database (Enverus DrillingInfo, 2020) together with EPA national totals for O/G production, gas processing, transmission, and distribution (EPA, 2020). Emissions from the oil and gas are assumed to be constant throughout each year with no seasonality. Prior anthropogenic emission totals for the continental US and Canada are 31.7 Tg a$^{-1}$ with major contributions from livestock (10.7 Tg a$^{-1}$), oil and gas (8.6 Tg a$^{-1}$ with 7.0 Tg a$^{-1}$ for the US and 1.6 Tg a$^{-1}$ for Canada), landfills (6.4 Tg a$^{-1}$), coal (3.2 Tg a$^{-1}$), wastewater treatment (0.72 Tg a$^{-1}$) and others. Prior wetland emissions with 0.5°×0.5° spatial resolution for individual months are taken from the mean of the nine highest-performance members of WetCHARTS v1.3.1 inventory ensemble (Ma et al., 2021). Fig. S4 shows the distribution of the prior methane emissions over the study domain. Although the EPA and ECCC bottom-up inventories report oil and gas emissions separately, spatial overlap between the two makes specific attribution difficult and we therefore combine them here.

**2.3 GEOS-Chem forward model simulations and inverse model setup**

We use the GEOS-Chem 12.7.0 chemical transport model (https://doi.org/10.5281/zenodo.1343546) as the forward model to relate methane emissions to the atmospheric methane columns observed by TROPOMI. GEOS-Chem is driven by GEOS-FP reanalysis meteorological fields from the NASA Global Modeling and Assimilation Office (GMAO) (Lucchesi, 2013) with 0.25°×0.3125° resolution. Here we use a nested version of GEOS-Chem with 0.25°×0.3125° horizontal resolution and dynamic boundary conditions from a 4°×5° global simulation. Following Shen et al. (2021), we correct the local boundary conditions on a daily basis by scaling to the ratio of TROPOMI and GEOS-Chem columns averaged over the neighboring ±1,000 km and ± 15 days.

The inversion optimizes a state vector defined by gridded methane emissions for the domain of interest. This involves constructing a Jacobian matrix $K$ that describes the sensitivity of model XCH4 to each emission state vector element. The construction is done by conducting sensitivity simulations in GEOS-Chem for the inversion period perturbing individual state vector elements in turn, and this is readily done on a high-performance cluster as a massively parallel problem. To reduce the computational cost, we limit the high-resolution inversion to the five domains where the O/G sources in the US and Canada are concentrated (Fig. 1 and more details in Fig. S5). These five domains encompass over 98% of total O/G emissions, 97% of oil production, and 99% of gas production in the continental US and Canada according to bottom-up information (Enverus DrillingInfo, 2020; Maasakkers et al., 2016; Scarpelli et al., 2021).

In each domain, we construct the state vector as follows. First, all native 0.25°×0.3125° grid cells with prior O/G emissions >0.5 Gg a$^{-1}$ are treated as independent state vector elements. These grid cells account for 93% of total O/G emissions in the US and Canada. Second, we aggregate grid cells that are inside each domain but with O/G emissions <0.5 Gg a$^{-1}$ into clusters using a k-means algorithm based on adjacency and consistency in prior sectoral emissions, following Turner and Jacob (2015). The average size of these clusters is 1°×1° and they allow us to retrieve another 5% of total O/G emissions in the US and Canada,

adding up to 98%. Third, we aggregate grid cells that are outside each domain but are within 4° in distance into 16 clusters using the k-means algorithm based on adjacency (Fig. S6), following Shen et al. (2021). These clusters are designed to correct for errors in boundary conditions, and they are not used for O/G source attribution. Altogether, the model estimates 3650 independent flux variables in the United States and Canada (Table S2). Methane sinks from oxidation and uptake by soils are included in GEOS-Chem but we do not optimize them here since they are irrelevant in nested model simulations where the loss of methane is by ventilation outside the domain (Varon et al., 2022).

## 2.4 Atmospheric inverse analysis

We solve for the posterior estimates of methane emissions (state vector $x$) in the US and Canada using Bayesian inverse analysis with Gaussian error statistics. The inversion finds the optimal estimate of $x$ by minimizing the cost function $J$ given by

$$J(x) = (x - x_A)^T S_A^{-1}(x - x_A) + \gamma(y - Kx)^T S_O^{-1}(y - Kx) \tag{2}$$

where $x_A$ is the prior estimate, $K$ is the Jacobian matrix, $y$ is the vector of TROPOMI observations, $S_A$ and $S_O$ are covariance matrices for prior and observational errors, and $\gamma$ is an additional regularization factor (Brasseur, and Jacob, 2017). The relationship between emissions and methane concentration (XCH4) is strictly linear since the sinks are not optimized (Varon et al., 2022). We construct the observational error covariance matrix $S_O$ by applying the residual error method, which assumes that the statistics of residual error (after removing the mean bias) between the observations and a GEOS-Chem simulation with prior emissions defines the observational error variance (Heald et al., 2004; Wecht et al., 2014). Both $S_A$ and $S_O$ are taken as diagonal, and we use $\gamma$ to avoid overfitting. For native grid cells, we assume 50% error standard deviation for all anthropogenic and natural emissions on the 0.25°×0.3125° grid. For the gridcell clusters, we assume the error standard deviation to be $\frac{50\%}{\sqrt{p}}$, where $p$ is the number of grid cells in each cluster.

The analytical solution for $\nabla_x J(x) = 0$ yields the optimal estimate $\hat{x}$ for the state vector, the corresponding posterior error covariance matrix $\hat{S}$, and the averaging kernel matrix $A$ as follows

$$\hat{x} = x_A + \left(\gamma K^T S_O^{-1} K + S_A^{-1}\right)^{-1} \gamma K^T S_O^{-1}(y - Kx_A) \tag{3}$$

$$\hat{S}^{-1} = \gamma K^T S_O^{-1} K + S_A^{-1} \tag{4}$$

$$A = I - \hat{S} S_A^{-1} \tag{5}$$

where $I$ is the identity matrix. The averaging kernel matrix $A$ defines the sensitivity of the posterior solution to the true state, and the diagonal terms of $A$ are the averaging kernel sensitivities diagnosing the ability of the inversion to quantify emissions for the corresponding state vector elements independently of the prior estimates. The trace of $A$ quantifies the degrees of freedom for signal (DOFS), representing the number of independent pieces of information that can be effectively optimized in the inversion (Brasseur, and Jacob, 2017).

The regularization term $\gamma$ is intended to account for unresolved observational error covariances in the inversion and thus to avoid overfit to observations. Following Lu et al. (2021), we choose $\gamma$ such that $(\hat{x} - x_A)^T S_A^{-1}(\hat{x} - x_A) \approx n$ where $n$ is the number of state vector elements, as would be expected from a chi-square distribution with $n$ degrees of freedom. This yields $\gamma$ in the range 0.1-0.4 with a best estimate of 0.2 (Fig. S7). We previously found a similar range of $\gamma$ using the L-curve method in a previous regional inversion of TROPOMI data for eastern Mexico (Shen et al., 2021).

We evaluate the inversion by comparing the column-averaged methane from TROPOMI with GEOS-Chem simulations using prior and posterior estimates (Fig. S8). The prior simulation has a negative bias of 10-15 ppb across most O/G basins and a positive bias of 10-20 ppb in the central and eastern US. GEOS-Chem simulations based on posterior estimates (as shown in Fig. 2) can reduce the negative bias to 0-10 ppb in most O/G basins and especially in the southwestern US where the TROPOMI observation frequency is high(Fig. S2 and S9).

**2.5 Partitioning the oil and natural gas emissions.**

Following Shen et al. (2021), we write the sectorial posterior correction for each grid cell as

$$f_i = \frac{\eta \alpha_i \sigma_{i,nation}^2 (f_0 - 1)}{\sigma_0^2} + 1 \qquad (1 \leq i \leq M) \tag{6}$$

$$\eta = \frac{\sigma_0^2}{\sum_{i=1}^{M} \alpha_i^2 \sigma_{i,nation}^2} \tag{7}$$

where $\alpha_i$ is the local fraction of emissions of each sector $i$ taken from the prior and $f_i$ is the posterior correction factor for that sector in this gridcell, $f_0$ is the posterior scaling factor for that state vector element, $\sigma_0$ is the prior error standard deviation, $M$ is the number of source sectors, $and$ $\sigma_{i,nation}$ refers to the error standard deviations on the national totals obtained from Maasakkers et al. (2016) and Bloom et al. (2017). The posterior correction factor will be adjusted more for a specific sector if this sector has higher percentage in the prior emissions and higher prior uncertainty.

**2.6 Inversion ensemble and uncertainty analysis**

Starting from the baseline inversion as described above, we conducted an ensemble of sensitivity inversions to test the robustness of our results to different inversion parameters and selection of TROPOMI observations. The 24-member ensemble

includes: (1) setting the regularization factor $\gamma$ to 0.1 and 0.4 (0.2 in baseline); (2) increase the prior O/G emissions by 50% (EPA and ECCC in baseline); (3) prior error standard deviation of 75% (50% in baseline); (4) removal of TROPOMI data with shortwave infrared albedo <0.05 (12% of total TROPOMI observations). The posterior covariance matrix $\hat{S}$ describes the error within the choice of each set of inversion parameters, and the ensemble allows us to explore the uncertainty arising from the selection of these inversion parameters. We use the Monte Carlo method to estimate the posterior uncertainty from the

ensemble. For each of the 24 members, we generate 100 samples from the posterior distribution, which yields 2,400 samples in total for each grid cell. We report error statistics on the inversion results as two standard deviations (2σ), corresponding to the 95% confidence level.

### 3 Quantification of oil and natural gas emissions in the US and Canada using TROPOMI

Fig. 1a shows the spatial distribution of TROPOMI column-averaged dry-air mole fraction of methane (XCH4) (Lorente et al., 2021) in the US and Canada from May 2018 to February 2020. The data shown in Fig. 1a are corrected for topography following Kort et al. (2014), but this correction is not used in the inverse analysis because the GEOS-Chem forward model accounts for topography. The largest values are along the southeastern coastal areas and the Mississippi River where wetland emissions are the dominant source (Fig. S10). Anthropogenic enhancements are also apparent in O/G basins including the

Central Valley in California, the Permian Basin, the Anadarko Basin, the Dallas Ft. Worth – Barnett Shale area, and southwestern Pennsylvania. Fig. 1b shows the bottom-up O/G methane emissions from the US and Canada in 2018 based on the gridded versions of the EPA and ECCC national inventories(Maasakkers et al., 2016; Scarpelli et al., 2021), which are used as prior estimates in our inversion framework. Here the original gridding of US EPA emissions for the year 2012 (Maasakkers et al., 2016) has been extrapolated to 2018 on the basis of the updated national inventory (EPA, 2020) and updated

information about O/G wells (Enverus DrillingInfo, 2020).

    Fig. 2a shows the optimized posterior correction factors for O/G emissions relative to the EPA and ECCC inventories (Fig. 1b). Fig. 2b shows the corresponding posterior emissions, and Fig. 2c shows the results for the 19 O/G basins (more details can be found in Fig. S11). Although the national maps show patterns of upward and downward correction factors, emissions for the 19 O/G basins show general increases except for parts of the Marcellus basin in southwestern Pennsylvania, California's

Central Valley and Denver-Julesburg basin. Emissions are dominated by a small number of basins where the correction factors to the national inventories are in excess of 2, except for the Marcellus. The Permian basin is the largest basin-wide source (2.9 Tg a$^{-1}$), a factor of 4.7 larger than the 0.62 Tg a$^{-1}$ in the extrapolated gridded EPA inventory, and accounts for 25% of total US O/G emissions in the posterior estimate. The average posterior uncertainty is 20% (2σ) for the first 9 largest O/G basins and 34% (2σ) for the 10 smaller ones, indicating that TROPOMI can more effectively quantify the emissions from the larger basins.

The underestimate of emissions by the gridded EPA inventory in the Permian has been pointed out before using satellite observations including TROPOMI (Zhang et al., 2020), GOSAT(Maasakkers et al., 2021), point source imagers (Irakulis-

Loitxate et al., 2021), and field studies(Lyon et al., 2021), and attributed in part to rapidly increasing oil and gas production (Zhang et al., 2020). Increasing our prior estimate of emissions in the Permian from 0.62 Tg a$^{-1}$ to 2.2 Tg a$^{-1}$ to reflect this knowledge increases our posterior estimate by 30% to 3.7 Tg a$^{-1}$ (Fig. 2c, more details in Fig. S12), a relatively small response reflecting the strong information available from TROPOMI observations. Splitting the TROPOMI observations into two periods, and with prior estimates of 0.62-2.2 Tg a$^{-1}$, we find posterior emissions of 2.5-3.4 Tg a$^{-1}$ for May 2018 - March 2019 and 3.0-3.8 Tg a$^{-1}$ for April 2019 - February 2020, indicating an increase over the period. The O/G emissions in the Delaware subbasin increase from 0.83 to 0.97 Tg a$^{-1}$, in response to the changes of prior emissions from 0.12 to 0.45 Tg a$^{-1}$. Assuming an average methane content of 80% for this natural gas, our posterior emission range of 2.9-3.7 Tg a$^{-1}$ corresponds to a 3.5-4.6% loss rate (natural gas production in the Permian was 5.4 x10$^6$ MMcf from May 2018 to February 2020 (Enverus DrillingInfo, 2020)).

We derive national totals for O/G emissions in the US and Canada by aggregating the posterior emissions from Fig. 2 and retaining the prior EPA and ECCC estimates for 0.2 Tg a$^{-1}$ of O/G emissions outside the inversion domains (including Alaska). Fig. 3 compares our results to previous studies, most of which are for emissions before 2017. Our satellite-derived US estimate in 2018-2020 is 12.6 ± 2.1(±2σ) Tg a$^{-1}$, which is 80% higher than the bottom-up inventory reported by EPA (EPA, 2020) and the Emissions Database for Global Atmospheric Research (EDGAR version v6.0) (Crippa et al., 2020). About 70% of this underestimate is from five O/G basins, including the Permian, Haynesville, Anadarko, Eagle Ford and Barnett, which are in total responsible for 40% of US O/G emissions (Fig. 2). Our US national estimate is comparable to the facility-based estimate for 2015 by Alvarez et al. (2018) that can better account for the heavy-tailed emissions and was found to be consistent with aircraft measurements. We find lower emissions than Alvarez et al. (Alvarez et al., 2018) in Denver-Julesburg, Fayetteville, Uinta, West Arkoma, San Juan and Northeast Pennsylvania, which could be due to decreasing O/G production in these basins ( Fig. S13). This is offset by fast growing emissions in the Permian, where the O/G production almost doubled from 2015 to 2019 (Enverus DrillingInfo, 2020; Zhang et al., 2020). Our US national estimate for O/G emissions is also comparable to previous inversions of GOSAT and in-situ data for 2010-2017 (Maasakkers et al., 2021; Lu et al., 2022), with Lu et al. (2022) reporting increasing emissions in the oil-producing basins but decreasing emissions in gas-producing basins over the period. When normalized by annual natural gas production (4.1x10$^7$ MMcf, US EIA; assuming the average CH$_4$ content is 80%) in 2019, the national O/G mean leakage rate (including all O/G sectors) inferred from our work is 2.0% in the US.

Our top-down estimate in Canada is 2.2 ± 0.6 (±2σ) Tg a$^{-1}$, which is 40% higher than the most recent ECCC reported emissions (ECCC, 2020) and EDGAR v6 (Crippa et al., 2020) in 2018, and is at the lower end of other top-down studies (2.3-3.6 Tg a$^{-1}$) for 2010-2017(Baray et al., 2021; Maasakkers et al., 2021; Chan et al., 2020; Lu et al., 2022). When normalized by annual natural gas production (7.1x10$^6$ MMcf, UNFCC https://unfccc.int/documents/271492; assuming the average CH$_4$ content is 90%) in 2019, the national O/G mean leakage rate is 1.8% in Canada.

We also calculated posterior emissions from the O/G sector using TROPOMI observations in different seasons. Here the prior O/G emissions remain constant over different seasons, so any changes in the posterior correction factors are determined by the inversion of the satellite data. Overall, the spatial distributions of posterior correction factors in spring, summer and autumn are consistent with that using the year-round data, especially in the south where TROPOMI observation density is high (Fig. S2). The posterior corrections from using wintertime data are slightly different in Canada and Northeastern US because of the low observation density and low averaging kernel sensitivities (Figure S14).

## 4 Comparison with field estimates for individual basins

A unique feature of our work is the use of satellite observations to quantify emissions at high resolution for individual O/G basins, building up to the national scale for the US and Canada. A number of aircraft and ground-based field campaigns previously estimated emissions from individual basins (Table S1). These field campaigns were carried out between 2013 and 2020 (with many of those field measurements taken before 2015), whereas our satellite observational period is for 2018-2020, which could affect the comparison. Intermittency of emissions is another factor that would affect the interpretation of results from field campaigns (Cusworth et al., 2021; Varon et al., 2021). Fig. 4 compares the basin-scale emission estimates from these field campaigns to the gridded EPA and ECCC inventories, and to our TROPOMI inversion results. The inventories are consistently lower by a factor of 1.5-3. Results from our TROPOMI inversion are more consistent with the field campaigns, especially for the large O/G basins ($>0.5$ Tg a$^{-1}$) such as the Permian, its Delaware sub-basin, Haynesville, and Barnett. The correlation coefficient ($R$) between TROPOMI-derived posterior estimates and field measurements is 0.96, compared to 0.85 for the EPA and ECCC inventories. Overall, the results offer quantitative support of our TROPOMI inversion for high-emitting basins ($>0.5$ Tg a$^{-1}$) but suggest that TROPOMI measurements provide only a limited constraint on the emission quantification of lesser-emitting basins. We examine below more broadly the parameters governing the capability of TROPOMI to quantify basin-scale emissions.

## 5 Assessing the quantification efficacy of TROPOMI for oil/gas basins

TROPOMI is designed to quantify emissions on a regional scale, and a critical question for methane emission controls is whether it can do so at the scale of individual O/G basins. Results in Fig. 2c and 4 indicate successful quantification of basin-scale emissions exceeding 0.5 Tg a$^{-1}$. The inherent TROPOMI limitations can be understood by examining the averaging kernel (AK) sensitivities of our inversion system. The AK sensitivities (diagonal terms of the AK matrix) measure the ability of the inversion to quantify the true emissions independently from the prior estimate (1 = fully, 0 = not at all). Our inversion assumes fixed (50%) prior emission error standard deviation on the $0.25° \times 0.3125°$ grid, so that absolute prior errors scale with the magnitude of emissions and decrease with the size of the basin. On the other hand, observational errors as estimated by the residual error method (Heald et al., 2004) generally remain in the 10-20 ppb range for individual observations and decrease with the number of observations. It follows that the ability of our TROPOMI inversion to quantify basin-scale emissions

increases with the magnitude of emissions and with the number of observations. The observation density is highest in the southwestern US where TROPOMI retrievals are most often successful (arid regions, clear skies, homogeneous surfaces), and thus the AK sensitivities are highest for O/G fields in those regions and in particular for the Permian (Fig. S2 and S9). We also estimate the relative error reduction from prior to posterior estimates, and our results show that the uncertainty decreases by an average of 40% (0-80%) across the 19 O/G basins (Fig. S15).

We examined more broadly the variables influencing the ability of our TROPOMI-based inversion system to quantify emissions at the basin scale. These variables include (a) emissions (prior and posterior emissions), (b) number of satellite observations, and (c) other geophysical properAlexe, M., Bergamaschi, P., Segers, A., Detmers, R., Butz, A., Hasekamp, O., Guerlet, S., Parker, R., Boesch, H., Frankenberg, C., Scheepmaker, R. A., Dlugokencky, E., Sweeney, C., Wofsy, S. C., and Kort, E. A.: Inverse modelling of $CH_4$ emissions for 2010–2011 using different satellite retrieval products from GOSAT and SCIAMACHY, Atmos. Chem. Phys., 15, 113–133, https://doi.org/10.5194/acp-15-113-2015, 2015.

Alvarez, R. A., Zavala-Araiza, D., Lyon, D. R., Allen, D. T., Barkley, Z. R., Brandt, A. R., Davis, K. J., Herndon, S. C., Jacob, D. J., Karion, A., Kort, E. A., Lamb, B. K., Lauvaux, T., Maasakkers, J. D., Marchese, A. J., Omara, M., Pacala, S. W., Peischl, J., Robinson, A. L., Shepson, P. B., Sweeney, C., Townsend-Small, A., Wofsy, S. C., and Hamburg, S. P.: Assessment of methane emissions from the U.S. oil and gas supply chain, Science, eaar7204, https://doi.org/10.1126/science.aar7204, 2018.

Bergamaschi, P., Karstens, U., Manning, A. J., Saunois, M., Tsuruta, A., Berchet, A., Vermeulen, A. T., Arnold, T., Janssens-Maenhout, G., Hammer, S., Levin, I., Schmidt, M., Ramonet, M., Lopez, M., Lavric, J., Aalto, T., Chen, H., Feist, D. G., Gerbig, C., Haszpra, L., Hermansen, O., Manca, G., Moncrieff, J., Meinhardt, F., Necki, J., Galkowski, M., O'Doherty, S., Paramonova, N., Scheeren, H. A., Steinbacher, M., and Dlugokencky, E.: Inverse modelling of European $CH_4$ emissions during 2006–2012 using different inverse models and reassessed atmospheric observations, Atmos. Chem. Phys., 18, 901–920, https://doi.org/10.5194/acp-18-901-2018, 2018.

Brasseur, G. P. and Jacob, D. J.: Modeling of atmospheric chemistry, Cambridge University Press, 2017.

Cressot, C., Chevallier, F., Bousquet, P., Crevoisier, C., Dlugokencky, E. J., Fortems-Cheiney, A., Frankenberg, C., Parker, R., Pison, I., Scheepmaker, R. A., Montzka, S. A., Krummel, P. B., Steele, L. P., and Langenfelds, R. L.: On the consistency between global and regional methane emissions inferred from SCIAMACHY, TANSO-FTS, IASI and surface measurements, Atmos. Chem. Phys., 14, 577–592, https://doi.org/10.5194/acp-14-577-2014, 2014.

Crippa, M., Guizzardi, D., Muntean, M., Schaaf, E., Dentener, F., van Aardenne, J. A., Monni, S., Doering, U., Olivier, J. G. J., Pagliari, V., and Janssens-Maenhout, G.: Gridded emissions of air pollutants for the period 1970–2012 within EDGAR v4.3.2, Earth Syst. Sci. Data, 10, 1987–2013, https://doi.org/10.5194/essd-10-1987-2018, 2018.

Crippa, M., Guizzardi, D., Muntean, M., Schaaf, E., Lo Vullo, E., Solazzo, E., Monforti-Ferrario, F., Olivier, J., and Vignati, E.: EDGAR v6.0 Greenhouse Gas Emissions. European Commission, Joint Research Centre (JRC), 2021.

Duren, R. M., Thorpe, A. K., Foster, K. T., Rafiq, T., Hopkins, F. M., Yadav, V., Bue, B. D., Thompson, D. R., Conley, S.,

Colombi, N. K., Frankenberg, C., McCubbin, I. B., Eastwood, M. L., Falk, M., Herner, J. D., Croes, B. E., Green, R. O., and Miller, C. E.: California's methane super-emitters, Nature, 575, 180–184, https://doi.org/10.1038/s41586-019-1720-3, 2019.

EIA-Venezuela: Venezuelan crude oil production falls to lowest level since January 2003, 2022.

Etiope, G., Ciotoli, G., Schwietzke, S., and Schoell, M.: Gridded maps of geological methane emissions and their isotopic signature, Earth Syst. Sci. Data, 11, 1–22, https://doi.org/10.5194/essd-11-1-2019, 2019.

European Commission and USA: About the Global Methane Pledge, 2021.

Frankenberg, C., Thorpe, A. K., Thompson, D. R., Hulley, G., Kort, E. A., Vance, N., Borchardt, J., Krings, T., Gerilowski, K., Sweeney, C., Conley, S., Bue, B. D., Aubrey, A. D., Hook, S., and Green, R. O.: Airborne methane remote measurements

reveal heavy-tail flux distribution in Four Corners region, Proc Natl Acad Sci USA, 113, 9734–9739, https://doi.org/10.1073/pnas.1605617113, 2016.

Fung, I., John, J., Lerner, J., Matthews, E., Prather, M., Steele, L. P., and Fraser, P. J.: Three-dimensional model synthesis of the global methane cycle, J. Geophys. Res., 96, 13033, https://doi.org/10.1029/91JD01247, 1991.

Heald, C. L., Jacob, D. J., Jones, D. B. A., Palmer, P. I., Logan, J. A., Streets, D. G., Sachse, G. W., Gille, J. C., Hoffman, R.

320 N., and Nehrkorn, T.: Comparative inverse analysis of satellite (MOPITT) and aircraft (TRACE-P) observations to estimate Asian sources of carbon monoxide: COMPARATIVE INVERSE ANALYSIS, J. Geophys. Res., 109, https://doi.org/10.1029/2004JD005185, 2004.

Hoglund-Isaksson, L., Gómez-Sanabria, A., Klimont, Z., Rafaj, P., and Schöpp, W.: Technical potentials and costs for reducing global anthropogenic methane emissions in the 2050 timeframe –results from the GAINS model, Environ. Res. Commun., 2,

025004, https://doi.org/10.1088/2515-7620/ab7457, 2020.

Hu, H., Hasekamp, O., Butz, A., Galli, A., Landgraf, J., Aan de Brugh, J., Borsdorff, T., Scheepmaker, R., and Aben, I.: The operational methane retrieval algorithm for TROPOMI, Atmos. Meas. Tech., 9, 5423–5440, https://doi.org/10.5194/amt-9-5423-2016, 2016.

IPCC: Climate Change 2021: The Physical Science Basis. Contribution of Working Group I to the Sixth Assessment Report

of the Intergovernmental Panel on Climate Change [Masson-Delmotte, V., P. Zhai, A. Pirani, S. L. Connors, C. Péan, S. Berger, N. Caud, Y. Chen, L. Goldfarb, M. I. Gomis, M. Huang, K. Leitzell, E. Lonnoy, J. B. R. Matthews, T. K. Maycock, T. Waterfield, O. Yelekçi, R. Yu and B. Zhou (eds.)]., Cambridge University Press, 2021.

Jacob, D. J., Turner, A. J., Maasakkers, J. D., Sheng, J., Sun, K., Liu, X., Chance, K., Aben, I., McKeever, J., and Frankenberg, C.: Satellite observations of atmospheric methane and their value for quantifying methane emissions, Atmos. Chem. Phys., 16,

14371–14396, https://doi.org/10.5194/acp-16-14371-2016, 2016.

Jacob, D. J., Varon, D. J., Cusworth, D. H., Dennison, P. E., Frankenberg, C., Gautam, R., Guanter, L., Kelley, J., McKeever, J., Ott, L. E., Poulter, B., Qu, Z., Thorpe, A. K., Worden, J. R., and Duren, R. M.: Quantifying methane emissions from the global scale down to point sources using satellite observations of atmospheric methane, Gases/Remote Sensing/Troposphere/Chemistry (chemical composition and reactions), https://doi.org/10.5194/acp-2022-246, 2022.

Janssens-Maenhout, G., Crippa, M., Guizzardi, D., Muntean, M., Schaaf, E., Dentener, F., Bergamaschi, P., Pagliari, V.,

Olivier, J. G. J., Peters, J. A. H. W., van Aardenne, J. A., Monni, S., Doering, U., Petrescu, A. M. R., Solazzo, E., and Oreggioni, G. D.: EDGAR v4.3.2 Global Atlas of the three major greenhouse gas emissions for the period 1970–2012, Earth Syst. Sci. Data, 11, 959–1002, https://doi.org/10.5194/essd-11-959-2019, 2019.

Lauvaux, T., Giron, C., Mazzolini, M., d'Aspremont, A., Duren, R., Cusworth, D., Shindell, D., and Ciais, P.: Global assessment of oil and gas methane ultra-emitters, Science, 375, 557–561, https://doi.org/10.1126/science.abj4351, 2022.

Lorente, A., Borsdorff, T., Butz, A., Hasekamp, O., aan de Brugh, J., Schneider, A., Wu, L., Hase, F., Kivi, R., Wunch, D., Pollard, D. F., Shiomi, K., Deutscher, N. M., Velazco, V. A., Roehl, C. M., Wennberg, P. O., Warneke, T., and Landgraf, J.: Methane retrieved from TROPOMI: improvement of the data product and validation of the first 2 years of measurements, Atmos. Meas. Tech., 14, 665–684, https://doi.org/10.5194/amt-14-665-2021, 2021.

Lu, X., Jacob, D. J., Zhang, Y., Maasakkers, J. D., Sulprizio, M. P., Shen, L., Qu, Z., Scarpelli, T. R., Nesser, H., Yantosca, R. M., Sheng, J., Andrews, A., Parker, R. J., Boesch, H., Bloom, A. A., and Ma, S.: Global methane budget and trend, 2010–2017: complementarity of inverse analyses using in situ (GLOBALVIEWplus CH$_4$ ObsPack) and satellite (GOSAT) observations, Atmos. Chem. Phys., 21, 4637–4657, https://doi.org/10.5194/acp-21-4637-2021, 2021.

Ma, S., Worden, J. R., Bloom, A. A., Zhang, Y., Poulter, B., Cusworth, D. H., Yin, Y., Pandey, S., Maasakkers, J. D., Lu, X., Shen, L., Sheng, J., Frankenberg, C., Miller, C. E., and Jacob, D. J.: Satellite Constraints on the Latitudinal Distribution and Temperature Sensitivity of Wetland Methane Emissions, AGU Advances, 2, https://doi.org/10.1029/2021AV000408, 2021.

Maasakkers, J. D., Jacob, D. J., Sulprizio, M. P., Scarpelli, T. R., Nesser, H., Sheng, J.-X., Zhang, Y., Hersher, M., Bloom, A. A., Bowman, K. W., Worden, J. R., Janssens-Maenhout, G., and Parker, R. J.: Global distribution of methane emissions, emission trends, and OH concentrations and trends inferred from an inversion of GOSAT satellite data for 2010–2015, Atmos. Chem. Phys., 19, 7859–7881, https://doi.org/10.5194/acp-19-7859-2019, 2019.

Maasakkers, J. D., Jacob, D. J., Sulprizio, M. P., Scarpelli, T. R., Nesser, H., Sheng, J., Zhang, Y., Lu, X., Bloom, A. A., Bowman, K. W., Worden, J. R., and Parker, R. J.: 2010–2015 North American methane emissions, sectoral contributions, and trends: a high-resolution inversion of GOSAT observations of atmospheric methane, Atmos. Chem. Phys., 21, 4339–4356, https://doi.org/10.5194/acp-21-4339-2021, 2021.

Oil & Gas Climate Initiative: Oil & Gas Climate Initiative Reporting Framework, 2021.

O'Rourke, P., Smith, Steven J, Mott, Andrea, Ahsan, Hamza, McDuffie, Erin E, Crippa, Monica, Klimont, Zbigniew, McDonald, Brian, Wang, Shuxiao, Nicholson, Matthew B, Feng, Leyang, and Hoesly, Rachel M.: CEDS v_2021_02_05 Release Emission Data (v_2021_02_05), https://doi.org/10.5281/ZENODO.4509372, 2021.

Pandey, S., Gautam, R., Houweling, S., van der Gon, H. D., Sadavarte, P., Borsdorff, T., Hasekamp, O., Landgraf, J., Tol, P., van Kempen, T., Hoogeveen, R., van Hees, R., Hamburg, S. P., Maasakkers, J. D., and Aben, I.: Satellite observations reveal extreme methane leakage from a natural gas well blowout, Proc Natl Acad Sci USA, 116, 26376–26381, https://doi.org/10.1073/pnas.1908712116, 2019.

Parker, R. J., Webb, A., Boesch, H., Somkuti, P., Barrio Guillo, R., Di Noia, A., Kalaitzi, N., Anand, J. S., Bergamaschi, P., Chevallier, F., Palmer, P. I., Feng, L., Deutscher, N. M., Feist, D. G., Griffith, D. W. T., Hase, F., Kivi, R., Morino, I., Notholt,

375   J., Oh, Y.-S., Ohyama, H., Petri, C., Pollard, D. F., Roehl, C., Sha, M. K., Shiomi, K., Strong, K., Sussmann, R., Té, Y., Velazco, V. A., Warneke, T., Wennberg, P. O., and Wunch, D.: A decade of GOSAT Proxy satellite CH$_4$ observations, Earth Syst. Sci. Data, 12, 3383–3412, https://doi.org/10.5194/essd-12-3383-2020, 2020.

Qu, Z., Jacob, D. J., Shen, L., Lu, X., Zhang, Y., Scarpelli, T. R., Nesser, H., Sulprizio, M. P., Maasakkers, J. D., Bloom, A.

380   A., Worden, J. R., Parker, R. J., and Delgado, A. L.: Global distribution of methane emissions: a comparative inverse analysis of observations from the TROPOMI and GOSAT satellite instruments, Atmos. Chem. Phys., 21, 14159–14175, https://doi.org/10.5194/acp-21-14159-2021, 2021.

Sadavarte, P., Pandey, S., Maasakkers, J. D., Lorente, A., Borsdorff, T., Denier van der Gon, H., Houweling, S., and Aben, I.: Methane Emissions from Superemitting Coal Mines in Australia Quantified Using TROPOMI Satellite Observations, Environ.

Sci. Technol., 55, 16573–16580, https://doi.org/10.1021/acs.est.1c03976, 2021.

Saunois, M., Bousquet, P., Poulter, B., Peregon, A., Ciais, P., Canadell, J. G., Dlugokencky, E. J., Etiope, G., Bastviken, D., Houweling, S., Janssens-Maenhout, G., Tubiello, F. N., Castaldi, S., Jackson, R. B., Alexe, M., Arora, V. K., Beerling, D. J., Bergamaschi, P., Blake, D. R., Brailsford, G., Brovkin, V., Bruhwiler, L., Crevoisier, C., Crill, P., Covey, K., Curry, C., Frankenberg, C., Gedney, N., Höglund-Isaksson, L., Ishizawa, M., Ito, A., Joos, F., Kim, H.-S., Kleinen, T., Krummel, P.,

Lamarque, J.-F., Langenfelds, R., Locatelli, R., Machida, T., Maksyutov, S., McDonald, K. C., Marshall, J., Melton, J. R., Morino, I., Naik, V., O'Doherty, S., Parmentier, F.-J. W., Patra, P. K., Peng, C., Peng, S., Peters, G. P., Pison, I., Prigent, C., Prinn, R., Ramonet, M., Riley, W. J., Saito, M., Santini, M., Schroeder, R., Simpson, I. J., Spahni, R., Steele, P., Takizawa, A., Thornton, B. F., Tian, H., Tohjima, Y., Viovy, N., Voulgarakis, A., van Weele, M., van der Werf, G. R., Weiss, R., Wiedinmyer, C., Wilton, D. J., Wiltshire, A., Worthy, D., Wunch, D., Xu, X., Yoshida, Y., Zhang, B., Zhang, Z., and Zhu, Q.:

The global methane budget 2000–2012, Earth Syst. Sci. Data, 8, 697–751, https://doi.org/10.5194/essd-8-697-2016, 2016.

Saunois, M., Stavert, A. R., Poulter, B., Bousquet, P., Canadell, J. G., Jackson, R. B., Raymond, P. A., Dlugokencky, E. J., Houweling, S., Patra, P. K., Ciais, P., Arora, V. K., Bastviken, D., Bergamaschi, P., Blake, D. R., Brailsford, G., Bruhwiler, L., Carlson, K. M., Carrol, M., Castaldi, S., Chandra, N., Crevoisier, C., Crill, P. M., Covey, K., Curry, C. L., Etiope, G., Frankenberg, C., Gedney, N., Hegglin, M. I., Höglund-Isaksson, L., Hugelius, G., Ishizawa, M., Ito, A., Janssens-Maenhout,

G., Jensen, K. M., Joos, F., Kleinen, T., Krummel, P. B., Langenfelds, R. L., Laruelle, G. G., Liu, L., Machida, T., Maksyutov, S., McDonald, K. C., McNorton, J., Miller, P. A., Melton, J. R., Morino, I., Müller, J., Murguia-Flores, F., Naik, V., Niwa, Y., Noce, S., O'Doherty, S., Parker, R. J., Peng, C., Peng, S., Peters, G. P., Prigent, C., Prinn, R., Ramonet, M., Regnier, P., Riley, W. J., Rosentreter, J. A., Segers, A., Simpson, I. J., Shi, H., Smith, S. J., Steele, L. P., Thornton, B. F., Tian, H., Tohjima, Y., Tubiello, F. N., Tsuruta, A., Viovy, N., Voulgarakis, A., Weber, T. S., van Weele, M., van der Werf, G. R., Weiss, R. F.,

Worthy, D., Wunch, D., Yin, Y., Yoshida, Y., Zhang, W., Zhang, Z., Zhao, Y., Zheng, B., Zhu, Q., Zhu, Q., and Zhuang, Q.: The Global Methane Budget 2000–2017, Earth Syst. Sci. Data, 12, 1561–1623, https://doi.org/10.5194/essd-12-1561-2020, 2020.

Scarpelli, T. R., Jacob, D. J., Maasakkers, J. D., Sulprizio, M. P., Sheng, J.-X., Rose, K., Romeo, L., Worden, J. R., and

Janssens-Maenhout, G.: A global gridded (0.1° × 0.1°) inventory of methane emissions from oil, gas, and coal exploitation based on national reports to the United Nations Framework Convention on Climate Change, Earth Syst. Sci. Data, 12, 563–575, https://doi.org/10.5194/essd-12-563-2020, 2020.

Scarpelli, T. R., Jacob, D. J., Grossman, S., Lu, X., Qu, Z., Sulprizio, M. P., Zhang, Y., Reuland, F., Gordon, D., and Worden, J. R.: Updated Global Fuel Exploitation Inventory (GFEI) for methane emissions from the oil, gas, and coal sectors: evaluation with inversions of atmospheric methane observations, Atmos. Chem. Phys., 22, 3235–3249, https://doi.org/10.5194/acp-22-3235-2022, 2022.

Schneising, O., Buchwitz, M., Reuter, M., Vanselow, S., Bovensmann, H., and Burrows, J. P.: Remote sensing of methane leakage from natural gas and petroleum systems revisited, Atmos. Chem. Phys., 20, 9169–9182, https://doi.org/10.5194/acp-20-9169-2020, 2020.

Shen, L., Zavala-Araiza, D., Gautam, R., Omara, M., Scarpelli, T., Sheng, J., Sulprizio, M. P., Zhuang, J., Zhang, Y., Qu, Z., Lu, X., Hamburg, S. P., and Jacob, D. J.: Unravelling a large methane emission discrepancy in Mexico using satellite observations, Remote Sensing of Environment, 260, 112461, https://doi.org/10.1016/j.rse.2021.112461, 2021.

Shen, L., Gautam, R., Omara, M., Zavala-Araiza, D., Maasakkers, J., Scarpelli, T., Lorente, A., Lyon, D., Sheng, J., Varon, D., Nesser, H., Qu, Z., Lu, X., Sulprizio, M., Hamburg, S., and Jacob, D.: Satellite quantification of oil and natural gas methane emissions in the US and Canada including contributions from individual basins, Gases/Remote Sensing/Troposphere/Chemistry (chemical composition and reactions), https://doi.org/10.5194/acp-2022-155, 2022.

Turner, A. J. and Jacob, D. J.: Balancing aggregation and smoothing errors in inverse models, Atmos. Chem. Phys., 15, 7039–7048, https://doi.org/10.5194/acp-15-7039-2015, 2015.

Turner, A. J., Jacob, D. J., Wecht, K. J., Maasakkers, J. D., Lundgren, E., Andrews, A. E., Biraud, S. C., Boesch, H., Bowman, K. W., Deutscher, N. M., Dubey, M. K., Griffith, D. W. T., Hase, F., Kuze, A., Notholt, J., Ohyama, H., Parker, R., Payne, V. H., Sussmann, R., Sweeney, C., Velazco, V. A., Warneke, T., Wennberg, P. O., and Wunch, D.: Estimating global and North American methane emissions with high spatial resolution using GOSAT satellite data, Atmos. Chem. Phys., 15, 7049–7069, https://doi.org/10.5194/acp-15-7049-2015, 2015.

UNFCCC: Greenhouse Gas Inventory Data Interface, 2021.

Varon, D. J., McKeever, J., Jervis, D., Maasakkers, J. D., Pandey, S., Houweling, S., Aben, I., Scarpelli, T., and Jacob, D. J.: Satellite Discovery of Anomalously Large Methane Point Sources From Oil/Gas Production, Geophys. Res. Lett., 46, 13507–13516, https://doi.org/10.1029/2019GL083798, 2019.

Veefkind, J. P., Aben, I., McMullan, K., Förster, H., de Vries, J., Otter, G., Claas, J., Eskes, H. J., de Haan, J. F., Kleipool, Q., van Weele, M., Hasekamp, O., Hoogeveen, R., Landgraf, J., Snel, R., Tol, P., Ingmann, P., Voors, R., Kruizinga, B., Vink, R., Visser, H., and Levelt, P. F.: TROPOMI on the ESA Sentinel-5 Precursor: A GMES mission for global observations of the atmospheric composition for climate, air quality and ozone layer applications, Remote Sensing of Environment, 120, 70–83, https://doi.org/10.1016/j.rse.2011.09.027, 2012.

Wecht, K. J., Jacob, D. J., Frankenberg, C., Jiang, Z., and Blake, D. R.: Mapping of North American methane emissions with

high spatial resolution by inversion of SCIAMACHY satellite data, J. Geophys. Res. Atmos., 119, 7741–7756, https://doi.org/10.1002/2014JD021551, 2014.

van der Werf, G. R., Randerson, J. T., Giglio, L., van Leeuwen, T. T., Chen, Y., Rogers, B. M., Mu, M., van Marle, M. J. E., Morton, D. C., Collatz, G. J., Yokelson, R. J., and Kasibhatla, P. S.: Global fire emissions estimates during 1997–2016, Earth Syst. Sci. Data, 9, 697–720, https://doi.org/10.5194/essd-9-697-2017, 2017.

Wunch, D., Wennberg, P. O., Toon, G. C., Connor, B. J., Fisher, B., Osterman, G. B., Frankenberg, C., Mandrake, L., O'Dell, C., Ahonen, P., Biraud, S. C., Castano, R., Cressie, N., Crisp, D., Deutscher, N. M., Eldering, A., Fisher, M. L., Griffith, D.

W. T., Gunson, M., Heikkinen, P., Keppel-Aleks, G., Kyrö, E., Lindenmaier, R., Macatangay, R., Mendonca, J., Messerschmidt, J., Miller, C. E., Morino, I., Notholt, J., Oyafuso, F. A., Rettinger, M., Robinson, J., Roehl, C. M., Salawitch, R. J., Sherlock, V., Strong, K., Sussmann, R., Tanaka, T., Thompson, D. R., Uchino, O., Warneke, T., and Wofsy, S. C.: A method for evaluating bias in global measurements of CO<sub>2</sub> total columns from space, Atmos. Chem. Phys., 11, 12317–12337, https://doi.org/10.5194/acp-11-12317-2011, 2011.

Zavala-Araiza, D., Omara, M., Gautam, R., Smith, M. L., Pandey, S., Aben, I., Almanza-Veloz, V., Conley, S., Houweling, S., Kort, E. A., Maasakkers, J. D., Molina, L. T., Pusuluri, A., Scarpelli, T., Schwietzke, S., Shen, L., Zavala, M., and Hamburg, S. P.: A tale of two regions: methane emissions from oil and gas production in offshore/onshore Mexico, Environ. Res. Lett., 16, 024019, https://doi.org/10.1088/1748-9326/abceeb, 2021.

Zhang, Y., Gautam, R., Pandey, S., Omara, M., Maasakkers, J. D., Sadavarte, P., Lyon, D., Nesser, H., Sulprizio, M. P., Varon,

D. J., Zhang, R., Houweling, S., Zavala-Araiza, D., Alvarez, R. A., Lorente, A., Hamburg, S. P., Aben, I., and Jacob, D. J.: Quantifying methane emissions from the largest oil-producing basin in the United States from space, Sci. Adv., 6, eaaz5120, https://doi.org/10.1126/sciadv.aaz5120, 2020.

Zhang, Y., Jacob, D. J., Lu, X., Maasakkers, J. D., Scarpelli, T. R., Sheng, J.-X., Shen, L., Qu, Z., Sulprizio, M. P., Chang, J., Bloom, A. A., Ma, S., Worden, J., Parker, R. J., and Boesch, H.: Attribution of the accelerating increase in atmospheric methane

during 2010–2018 by inverse analysis of GOSAT observations, Atmos. Chem. Phys., 21, 3643–3666, https://doi.org/10.5194/acp-21-3643-2021, 2021.

ties and satellite retrieval parameters (e.g. the albedo, surface altitude, surface roughness). Of all these variables, emissions and the number of satellite observations show the strongest correlation with the posterior uncertainty for the 19 O/G basins; the correlation coefficient $R$ can be as high as -0.7, a result consistent with AK sensitivities (Fig. 5, more details in Fig. S16).

Results across these O/G basins show that our inversion framework can quantify area methane emissions with an average uncertainty below 30% if the emission rates exceed 0.2 Tg $a^{-1}$ and the number of observations exceeds 5,000 $a^{-1}$. If we normalize the number of observations by the basin area, it suggests that our inversion framework can quantify large basin-scale sources where the satellite data density is greater than 0.3 counts $km^{-2}$ $a^{-1}$ (Fig. 5). This encompasses many O/G fields at mid-latitudes, though O/G fields in the tropics are more of a challenge because they are often collocated with wetlands (Nigeria,

Venezuela) and therefore have extensive cloudiness (Fig. S17). For areas with lower data density, a reliable quantification may need the support of other observations (e.g., other satellites, field measurements) or more accurate facility-scale information.

From the data in Fig. 5, the basin-scale posterior relative uncertainty (%) of our inversion framework can be estimated using the following equation.

$$z = -15 \log_{10} x_1 - 13x_2 - 17 \quad (R^2=0.53) \tag{8}$$

where $z$ is the posterior relative uncertainty (%), $x_1$ is the bottom-up emission (in Tg a$^{-1}$) for the basin, and $x_2$ is the satellite data density (in counts km$^{-2}$ a$^{-1}$).

We further test this conclusion by examining 1,000 pseudo-basins that are generated randomly with varying locations and area sizes (Fig. S18) in the US and Canada. Unlike the 19 O/G basins that are usually located in arid regions with denser observations, these 1,000 pseudo-basins can encompass more complicated satellite observing conditions and sectorial emission

constitutions. As seen from Fig. S19, our inversion framework can constrain the posterior O/G emissions with an uncertainty <30% in areas with O/G emission rates > 0.2-0.5 Tg a$^{-1}$ and the number of observations is higher than 5x10$^3$ a$^{-1}$. Our result suggests that TROPOMI can be useful in assessing large area sources with emissions exceeding 0.2-0.5 Tg a$^{-1}$ and observation counts exceeding 5000 a$^{-1}$.

**6 Discussion**

In summary, we have shown that TROPOMI satellite observations can successfully quantify methane emissions from the oil and natural gas (O/G) sector in the US and Canada, and resolve the contributions from individual productionbasins. This involved inversions of TROPOMI observations for 22 months (May 2018 – February 2020) at 0.25°×0.3125° resolution in the O/G production basins and other O/G-emitting grid cells, accounting for over 98% of total O/G emissions in the continental US EPA and Canada ECCC national inventories used as prior estimates for the inversion. We conducted an ensemble of

inversions to determine the sensitivity of results to different weighting of observations, different prior estimates and associated uncertainties, and the addition of data quality filters. We find that national methane emissions from the O/G sector are 12.6 ± 2.1 (±2σ) Tg a$^{-1}$ in the US and 2.2 ± 0.6 (±2σ) Tg a$^{-1}$ in Canada, which are 80% and 40% higher than the national bottom-up inventories, respectively. About 70% of the discrepancy in the EPA inventory can be attributed to five O/G basins: the Permian, Haynesville, Anadarko, Eagle Ford and Barnett Basin, which in total account for 40% of US emissions.  Our satellite-derived

emission estimates show good consistency with in-situ field measurements for large O/G basins with emissions higher than 0.5 Tg a$^{-1}$. Further examination of the error budget of the inversion suggests that the TROPOMI observations can quantify emission rates with an uncertainty (2σ) better than 30% in areas with emissions exceeding 0.2-0.5 Tg a$^{-1}$ and observation counts exceeding 5000 a$^{-1}$. Many large O/G basins at mid-latitudes meet these criteria for successful source quantification.

**Acknowledgments**

L.S. is thankful to the High Meadows Research Fellowship at EDF for supporting this work. R.G., M.O., D.Z.A., D.L. and S.P.H. at EDF were funded by the Robertson Foundation. Work at Harvard was funded by the NASA Carbon Monitoring System and by the United Nations Environment Program. The TROPOMI data processing was carried out on the Dutch National e-Infrastructure with the support of the SURF Cooperative. All data, code, and materials used in the analyses is available at https://doi.org/10.18170/DVN/JPKFU6.

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

**Figures and Tables**

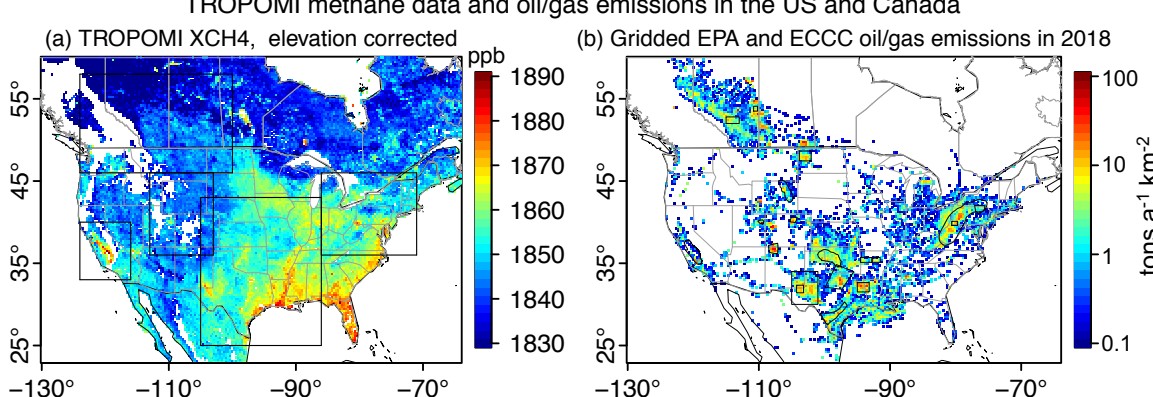

**Figure 1**. **TROPOMI methane observations and prior estimates of oil and natural gas emissions in the US and Canada.**
(a) TROPOMI satellite observations of column-averaged dry methane mixing ratio (XCH4) averaged from May 2018 to February 2020, mapped to 0.25°×0.3125° resolution and corrected for surface topography as 7 ppb/km (Kort et al., 2014). This elevation correction is only for visualization. We conduct the inversions for the five rectangular domains shown in black that account for over 98% of O/G emissions in the continental US and Canada. (b) Gridded national inventory emissions from the oil and natural gas sector in the US and Canada in 2018 used as prior estimates in our inversion of TROPOMI observations.
Grid cells with emission fluxes <0.1 tons a$^{-1}$ km$^{-2}$ are shown as white. The boundaries of the 19 major O/G basins are shown on the map and the names of these basins can be found in Fig. S11.

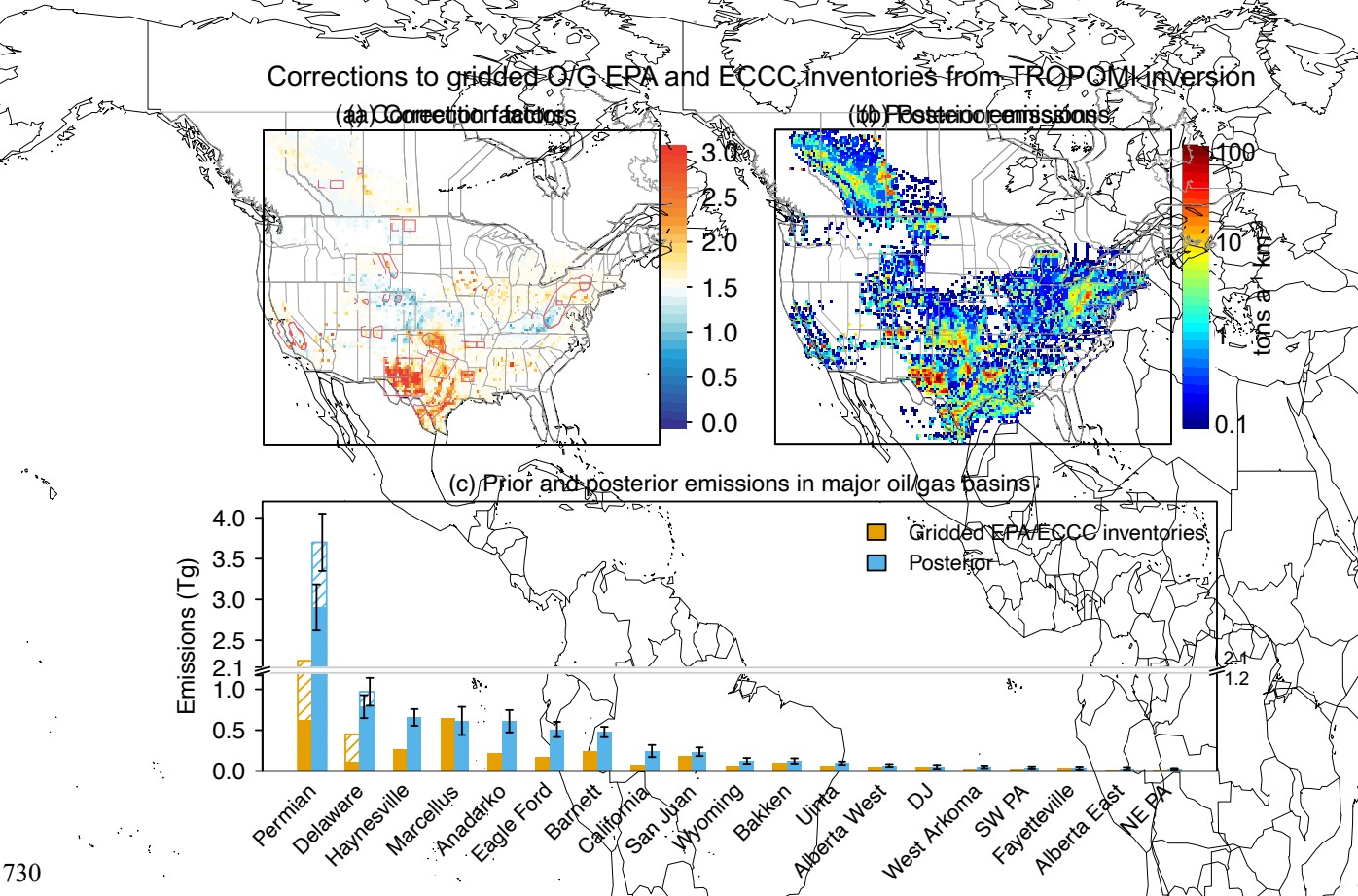

**Figure 2. Corrections to oil and natural gas methane emissions in the US EPA and Canada ECCC national inventories from the inversion of TROPOMI methane observations** (May 2018 – February 2020). (a) Posterior correction factors to the gridded national inventory estimates shown in Fig. 1b and used as prior estimates in the inversion. The boundaries of the 19 major O/G basins are shown on the map and the names of these basins can be found in Fig. S11. (b) Posterior O/G methane emissions. For (a) and (b), grid cells with prior O/G emissions <0.1 tons a$^{-1}$ km$^{-2}$ are shown as white (consistent with Fig. 1b). (c) Prior and posterior emissions in the 19 oil and gas basins, arranged in decreasing order of posterior emissions. Delaware is a sub-region of the Permian basin. Vertical bars indicate the 2x error standard deviations from the inversion ensemble. Striped bars for the Permian and Delaware basins show the results of an inversion where the prior estimate of emissions from the oil and gas production sector was increased by a factor of 4 from the EPA inventory, reflecting previous evidence that the EPA inventory is too low.

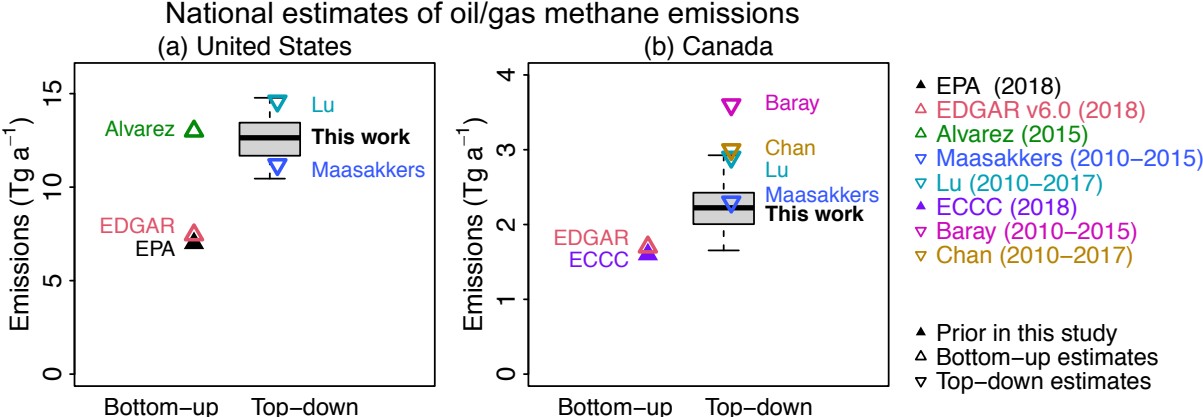

**Figure 3**. **Bottom-up and top-down estimates of national oil and natural gas methane emissions in the US and Canada**. Estimates from this work are shown as quantile plots for the inversion ensemble. The top and bottom of the box are the 25th and 75th percentile, the vertical bars are for the minimum and maximum, and the centre line is the 50th percentile. Mean emissions ±2 standard deviations from the inversion ensemble are $12.6 \pm 2.1$ Tg a$^{-1}$ for the US and $2.2 \pm 0.6$ Tg a$^{-1}$ for Canada. Symbols show previous estimates including EPA (2020) for 2018 (which equals the prior estimate for our work), EDGAR v6.0 (Crippa et al., 2020) for 2018, Alvarez et al. (2018) for 2015, Maasakkers et al. (2021) for 2010-2015, Lu et al. (2022) for 2010-2017, ECCC (2020) for 2018 (used as prior estimate for our work), Baray et al. (2021) for 2010-2015, and Chan et al. (2020) for 2010-2017. Maasakkers et al. (2021) and Lu et al. (2022) did not include the O/G emissions in Alaska so we add 0.1 Tg a$^{-1}$ of emissions (EPA, 2020; Maasakkers et al., 2016) here to obtain the US national total.

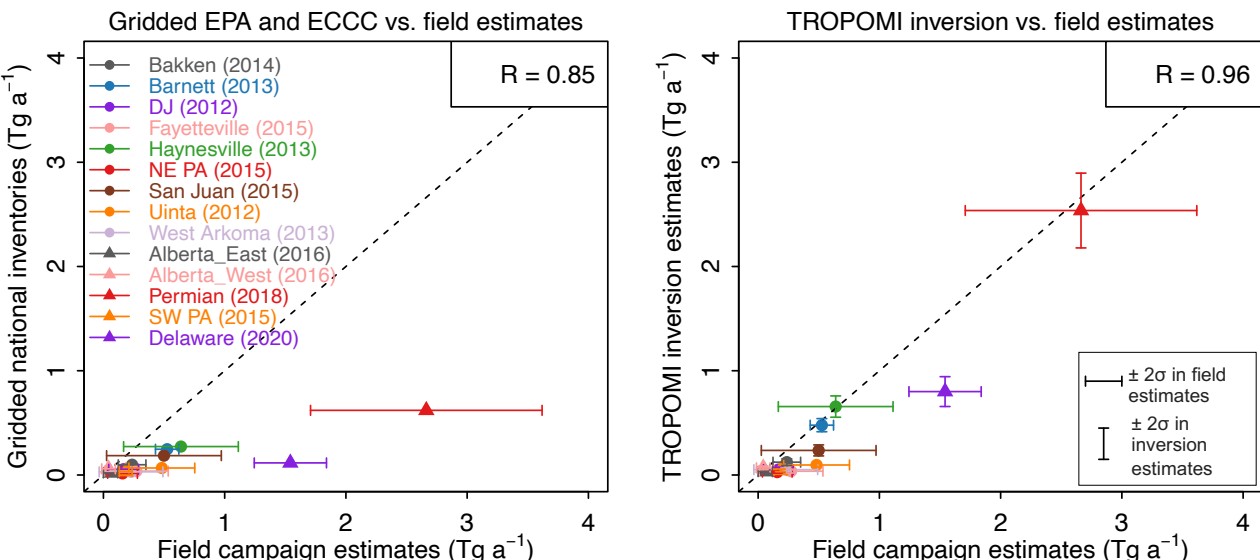

**Figure 4**. **Methane emissions from 14 oil and natural gas production basins in the US**. Estimates from field campaigns are compared to the gridded EPA and ECCC inventories for the US and Canada (left panel) and to results from our TROPOMI inversion using these inventories as prior estimates (right panel). The 1:1 line is dashed and correlation coefficient (R) are shown inset. More details including references for the field campaigns can be found in Table S1. Circles represent the same 9 basins in Alvarez et al. (2018) and triangles are the new basins included in this study.

760

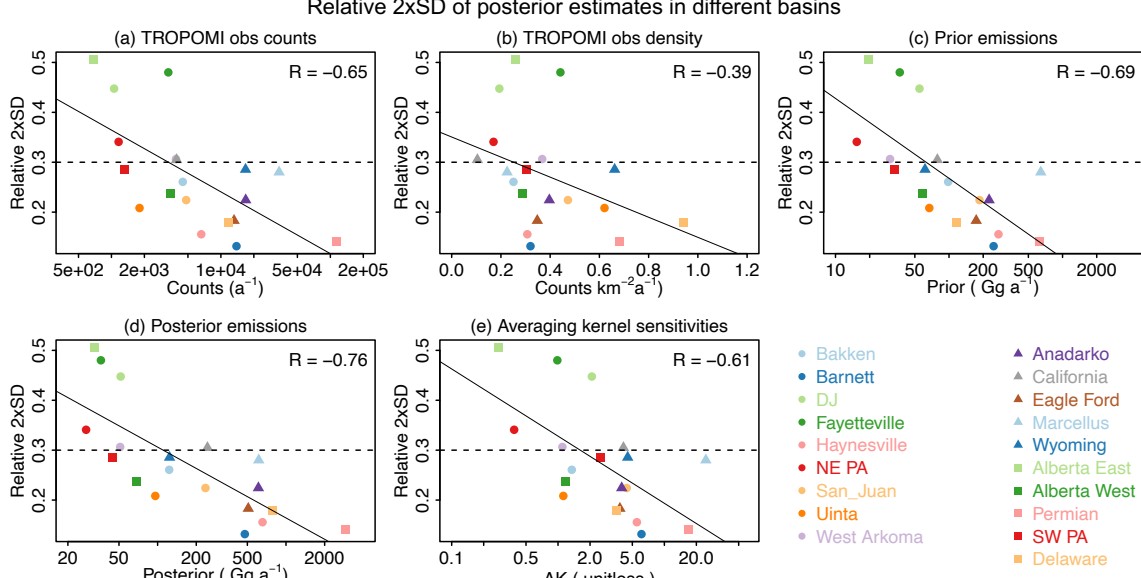

**Figure 5**. Relationship of relative standard deviation (we use 2σ here) of satellite-derived posterior estimates with different variables, including (a) the number of TROPOMI observations per year, (b) the satellite observation density, (c) prior emissions, (d) posterior emissions, and (e) averaging kernel sensitivities. The correlation coefficients are shown inset. The boundaries of 19 oil/gas basins are overlain on the map and the names of these basins can be found in Fig. S11.