# Peer review of "contributions from individual basins"

_Atmospheric Chemistry and Physics, 2022_

## Referee Comment (RC1)

In this paper, the authors use a high-resolution (~ 25 km) inverse modeling to estimate methane emissions from individual oil and natural gas (O/G) basins in the US and Canada based on 22-month satellite observations from TROPOMI. The authors compared their results with wildly-used "bottom-up" emission inventories and other "top-down" emissions. The authors also evaluated the uncertainties from the model and observations. The topic of the paper fits the scope of ACP, and it provides a way to quantify the O/G methane emissions. It is recommended to publish after the authors address the following aspects.

**General comments:**

1, Do the numbers of samplings in different seasons affect the estimated posterior emissions? The observations in the winter, especially over Canada, are limited because of the snow and high solar zenith angle. Did the authors evaluate the influence of uneven sampling in different seasons?

2, New infrastructures could also contribute to an increase in $CH_4$ emissions (e.g., in Permian Basin). These new sources, however, are not reported in a priori emission inventories. Besides, scaling a priori emissions to a certain year could not solve this problem, either. How did the authors deal with these "missing" emissions? Whether the model can correctly locate these emissions that are not in the emissions inventory?

3, Line 221 to 258: About the discussion over Canada, the authors argue that the lower emission than other "top-down" inventories is possibly due to a decreasing trend of O/G emissions after 2014 in Canada. It is quite tricky to argue in this way. The estimation in this paper is still 40% higher than that of ECCC-reported emissions (ECCC, 2020) and EDGAR v6. If the authors want to draw this conclusion, the authors should first prove both "top-down" inventories and "bottom-up" inventories catch the same trend but only show differences in absolute values.

4, The section 3 has a lot of repetitive content with the method section. Please combine them and reorganize the structure of the paper.

5, Line 310: I doubt the argument here. First, the number of observations of TROPOMI is limited by the retrieval over water. Many offshore oil/gas sources (e.g., the Middle East) are difficult to be resolved. Second, as shown in Fig S17, many places in the world have no data even with a 22-month recording.

**Specific comments:**

1, Line 49 and 56: Please check the format of the two references of Lu et al.

2, Line 64: Please give the definition of the blended albedo or refer to the relevant reference.

3, Line 121: The projects of Fig. S5 and S6 seem to be distorted. Please use right projections.

4, Line 122: "gridcells" should be "grid cells". Please correct all of them in the paper.

5, Line 126-127: How about the new sources? Although the emissions from bottom-up inventory can be scaled to the later years, the locations won't change, which means the new sources are not included.

6, Line 182: Please specify if the authors used $XCH_4$ with the surface correction. According to Figure 1, the authors also should clarify here that $XCH_4$ has been corrected by the elevation.

7, Line 196: "x" should be "×". Please check the paper and correct all of them.

8, Line 215: Any explanations about the decreases?

9, Line 295: A typo of "areal"?

---

## Author Comment (AC1)

**Response to referee comments on "Satellite quantification of oil and natural gas methane emissions in the US and Canada including contributions from individual basins"**

We thank the two referees for their careful reading of the manuscript and the valuable comments. This document is organized as follows: the Referee's comments are in *italic*, our responses are in plain text, and all the revisions in the manuscript are shown in blue. **Blue text** here denotes text written in direct response to the Referee's comments. The line numbers in this document refer to the updated **WORD** manuscript **with tracked changes**.

**Referee 1**

*In this paper, the authors use a high-resolution (~ 25 km) inverse modeling to estimate methane emissions from individual oil and natural gas (O/G) basins in the US and Canada based on 22-month satellite observations from TROPOMI. The authors compared their results with wildly-used "bottom-up" emission inventories and other "top-down" emissions. The authors also evaluated the uncertainties from the model and observations. The topic of the paper fits the scope of ACP, and it provides a way to quantify the O/G methane emissions. It is recommended to publish after the authors address the following aspects.*

**General comments:**

*1, Do the numbers of samplings in different seasons affect the estimated posterior emissions? The observations in the winter, especially over Canada, are limited because of the snow and high solar zenith angle. Did the authors evaluate the influence of uneven sampling in different seasons?*

**Response**. Thanks. We have a new supplementary figure to show the posterior correction factors from using TROPOMI data in different seasons.

Line 385. We also calculated posterior emissions from the O/G sector using TROPOMI observations in different seasons. Overall, the spatial distributions of posterior correction factors in spring, summer and autumn are consistent with that using the year-round data, especially in the south where TROPOMI observation density is high (Fig. S2). The posterior corrections from using wintertime data are slightly different in Canada and Northeastern US because of the low observation density and low averaging kernel sensitivities (Figure S14).

[Figure]

**Figure S14.** Posterior correction factors relative to the prior inventory and averaging kernel sensitivities using TROPOMI data in different seasons.

*2, New infrastructures could also contribute to an increase in CH4 emissions (e.g., in Permian Basin). These new sources, however, are not reported in a priori emission inventories. Besides, scaling a priori emissions to a certain year could not solve this problem, either. How did the authors deal with these "missing" emissions? Whether the model can correctly locate these emissions that are not in the emissions inventory?*

**Response**. Thanks for pointing this out. We have already considered new sources using the Enverus DrillingInfo database.

Line 101. We extrapolated the US emissions for the O/G production sector to 2018 based on upstream well data in the Enverus DrillingInfo database (Enverus DrillingInfo, 2020) together with EPA national totals for O/G production, gas processing, transmission, and distribution (EPA, 2020)

*3, Line 221 to 258: About the discussion over Canada, the authors argue that the lower emission than other "top-down" inventories is possibly due to a decreasing trend of O/G emissions after*

*2014 in Canada. It is quite tricky to argue in this way. The estimation in this paper is still 40% higher than that of ECCC-reported emissions (ECCC, 2020) and EDGAR v6. If the authors want to draw this conclusion, the authors should first prove both "top-down" inventories and "bottom-up" inventories catch the same trend but only show differences in absolute values.*

**Response**. This is a good point. We have removed this argument to make our narrative more accurate.

Line 297.

*4, The section 3 has a lot of repetitive content with the method section. Please combine them and reorganize the structure of the paper.*

**Response**. We have removed the 2nd and 3rd paragraphs in the old manuscript and combine these contents with the Methods

*5, Line 310: I doubt the argument here. First, the number of observations of TROPOMI is limited by the retrieval over water. Many offshore oil/gas sources (e.g., the Middle East) are difficult to be resolved. Second, as shown in Fig S17, many places in the world have no data even with a 22-month recording.*

**Response**. We now soften the arguments here. We deleted the argument that it can be used to effectively assess global O/G emissions. Now we say

Line 450. As seen from Fig. S19, our inversion framework can constrain the posterior O/G emissions with an uncertainty <30% in areas with O/G emission rates > 0.2-0.5 Tg $a^{-1}$ and the number of observations is higher than 5x10$^3$ $a^{-1}$. Our result suggests that TROPOMI can be useful in assessing large area sources with emissions exceeding 0.2-0.5 Tg $a^{-1}$ and observation counts exceeding 5000 $a^{-1}$.

**Specific comments:**

*1, Line 49 and 56: Please check the format of the two references of Lu et al.*

**Response.** We have updated the reference. Thanks.

*2, Line 64: Please give the definition of the blended albedo or refer to the relevant reference.*

**Response.** Now we say

Line 88. The blended albedo is a weighted difference of near-infrared (NIR) and SWIR albedos to filter scenes covered by snow(Wunch et al., 2011).

*3, Line 121: The projects of Fig. S5 and S6 seem to be distorted. Please use right projections.*

**Response.** We have updated Figure S6. Please check.

*4, Line 122: "gridcells" should be "grid cells". Please correct all of them in the paper.*

**Response.** Corrected throughout the text.

*5, Line 126-127: How about the new sources? Although the emissions from bottom-up inventory can be scaled to the later years, the locations won't change, which means the new sources are not included.*

**Response.** We have already considered the new sources using the Enverus DrillingInfo database when we scale the inventory to the year 2018.

Line 100. We extrapolated the US emissions for the O/G production sector to 2018 based on upstream well data in the Enverus DrillingInfo database (Enverus DrillingInfo, 2020) together with EPA national totals for O/G production, gas processing, transmission, and distribution (EPA, 2020)

*6, Line 182: Please specify if the authors used XCH4 with the surface correction. According to Figure 1, the authors also should clarify here that XCH4 has been corrected by the elevation.*

**Response.** Now we say:

Line 261. The data shown in Fig. 1a are corrected for topography following Kort et al. (2014), but this correction is not used in the inverse analysis because the GEOS-Chem forward model accounts for topography

*7, Line 196: "x" should be "×". Please check the paper and correct all of them.*

**Response.** Corrected.

*8, Line 215: Any explanations about the decreases?*

**Response.** Sorry, we don't know the reason. We guess it is related to more stringent emission control in these traditional O/G basins, but we don't find evidence to support this.

*9, Line 295: A typo of "areal"?*

**Response.** Corrected.

---

## Author Comment (AC2)

**Response to referee comments on "Satellite quantification of oil and natural gas methane emissions in the US and Canada including contributions from individual basins"**

We thank the two referees for their careful reading of the manuscript and the valuable comments. This document is organized as follows: the Referee's comments are in *italic*, our responses are in plain text, and all the revisions in the manuscript are shown in blue. Blue text here denotes text written in direct response to the Referee's comments. The line numbers in this document refer to the updated **WORD** manuscript with tracked changes.

**Referee 2**

This work by Shen et al. uses TROPOMI XCH4 retrievals to quantify oil/gas emissions at high GEOS-Chem resolution of 0.25 by 0.3125 degree. It follows the optimal estimation framework with GEOS-Chem as forward model developed in the TROPOMI-Permian study by Zhang et al. (2018), GOSAT/in-situ-global study by Lu et al. (2021), GOSAT/in-situ-North America study by Lu et al. (2022), and TROPOMI-Mexico study by Shen et al. (2021). This work also presents significantly improved spatial resolution of emission estimates and detailed basin-level emission comparison with previous quantification by airborne measurements. It is recommended for publication in ACP after addressing the following issues.

General comments:

Some languages are still reminiscence from previous version of the manuscript. For example, "see Methods for more details", whereas there is no "Methods" section. The first three paragraphs of section 3 is largely redundant.

**Response.** Thanks for pointing this out. We have removed the 2nd and 3rd paragraphs in our old manuscript and combine related contents with the Method part.

It is suggested to clarify and maybe expand section 2.6 on the ensemble uncertainty analysis, which seems to be an advance from previous GC analytical inversion studies from the group. How does this "posterior uncertainty" compare with the classic posterior uncertainty calculated from equation 4? Where does the number 2400 come from at line 178? Posterior emission at each grid cell drawn from Gaussian PDF 100 times and then multiplied by 24?

**Response.** Thanks for pointing this out. We have added these contents to clarify our ensemble uncertainty analysis.

Line 252. The posterior covariance matrix  $\hat{S}$  describes the error within the choice of each set of inversion parameters, and the ensemble allows us to explore the uncertainty arising from the selection of these inversion parameters. We use the Monte Carlo method to estimate the posterior uncertainty from the ensemble. For each of the 24 members, we generate 100 samples from the posterior distribution, which

yields 2,400 samples in total for each grid cell. We report error statistics on the inversion results as two standard deviations  $(2\sigma)$ , corresponding to the 95% confidence level.

Section 5 gives a novel way of quantifying TROPOMI's capacity to constrain basin-scale emissions. However, the posterior uncertainty threshold of 30% seems arbitrary and presumably closely related to the prior uncertainty assumed (50% here). Equation 8 may be misleading as it shows that the posterior relative uncertainty is driven by basin- total emissions and satellite coverage only. Consider adding prior uncertainty as a predictor, or using the relative reduction from prior error to posterior error.

**Response.** We tried to add prior uncertainty as a predictor, and we don't find it can significantly improve the performance of our linear regression model (Equation 8). This is because posterior uncertainty is less dependent on prior information when the satellite density is high. This also means our posterior uncertainty is not closely related to the prior uncertainty.

The referee has made a good point about the relative reduction of errors, and we add a new supplementary figure.

**Line 420.** We also estimate the relative error reduction from prior to posterior estimates, and our results show that the uncertainty decreases by an average of 40% (0-80%) across the 19 O/G basins (Fig. S15).

**Figure S15.** Ratio of posterior errors to prior errors for the 19 basins. The basin-scale prior errors are calculated using a similar approach as described in Section 2.6.

Specific comments:

Lines 112-115: is this the "specification of boundary conditions" mentioned in lines 91-92? It is not very clear what the "vertical fields" are, and how GC CH4 fields are corrected exactly. Was GC CH4 scaled every day, so that the mean of the boundary grids of the North America domain matches the mean of TROPOMI pixels within a buffer zone?

**Response.** Now we say this in text.

Line 127. Following Shen et al. (2021), we correct the local boundary conditions on a daily basis by scaling to the ratio of TROPOMI and GEOS-Chem columns averaged over the neighboring  $\pm 1,000$  km and  $\pm 15$  days.

*Line 122: it implies that the time variation of CH4 emission at each grid cell is not considered. Is constant emission assumed throughout the study period? Please clarify.*

Response. Now we make it clear in text.

Line 115. Emissions from the oil and gas are assumed to be constant throughout each year with not seasonality.

Line 135: this equation implies that the model is linear so the jacobian (K) can be calculated only once. A linear forward model is essential for this framework given the cost of generating K. It is suggested to add a reference or calculations to justify that GC CH4 simulation is linear.

**Response.** Thanks. Now we say this in text.

Line 131. This involves constructing a Jacobian matrix K that describes the sensitivity of model XCH4 to each emission state vector element. The construction is done by conducting sensitivity simulations in GEOS-Chem for the inversion period perturbing individual state vector elements in turn, and this is readily done on a high-performance cluster as a massively parallel problem.

Line 168. Methane sinks from oxidation and uptake by soils are included in GEOS-Chem but we do not optimize them here since they are irrelevant in nested model simulations where the loss of methane is by ventilation outside the domain (Varon et al., 2022).

Line 177. The relationship between emissions and methane concentration (XCH4) is strictly linear since the sinks are not optimized (Varon et al., 2022).

*Line 136: it is suggested to provide more information on how K and S\_o are constructed and how the computational challenges are solved, given the very large number of observations (7e6).*

**Response**. Here we make it clear in text.

Line 181. Both  $S_A$  and  $S_O$  are taken as diagonal, and we use  $\gamma$  to avoid overfitting.

Line 206. The regularization term  $\gamma$  is intended to account for unresolved observational error covariances in the inversion and thus to avoid overfit to observations.

*Line 154: suggest to comment on the rationale of choosing this over the L-curve approach in some previous works.*

Line 207. Following Lu et al. (2021), we choose  $\gamma$  such that  $(\hat{x} - x_A)^T S_A^{-1} (\hat{x} - x_A) \approx n$  where *n* is the number of state vector elements, as would be expected from a chi-square distribution with *n* degrees of freedom. This yields  $\gamma$  in the range 0.1-0.4 with a best estimate of 0.2 (Fig. S7). We previously found a

similar range of  $\gamma$  using the L-curve method in a previous regional inversion of TROPOMI data for eastern Mexico (Shen et al., 2021).

Line 169: it is "sigma i, nation" that refers to the error standard deviations.

**Response. Now we say.**

Line 223.  $\sigma_{i,nation}$  refers to the error standard deviations on the national totals obtained from Maasakkers et al. (2016) and Bloom et al. (2017).

Line 182: suggest emphasize that this is "topography-corrected" XCH4 in the text.

Response. Now we say.

Line 261. The data shown in Fig. 1a are corrected for topography following Kort et al. (2014), but this correction is not used in the inverse analysis because the GEOS-Chem forward model accounts for topography.

*Line 220: is 20% the sum of the first 9 largest O/G basins' uncertainty, or the mean of them?*

Response. Now we say:

Line 277. The average posterior uncertainty is 20% ( $2\sigma$ ) for the first 9 largest O/G basins

Line 269 and Figure 4: is R2 from the fitted ordinary least squares line between this work and inventory/field work? Different fitting lines may give different R2. Pearson correlation coefficient might be more proper here.

**Response**. Thanks for pointing this out. Both methods give very similar R2.

Line 279: update the "Methods" section.

**Response. Done.**

*Line 280-281: please clarify/confirm what is "error variances weighted by the corresponding error covariances".*

**Response**. Now we say.

Line 409. The AK sensitivities (diagonal terms of the AK matrix) measure the ability of the inversion to quantify the true emissions independently from the prior estimate (1 = fully, 0 = not at all).

---

## Referee Report (RR1)

Line 87: "$-3.4 \pm 5.6$" and "US (Lorente et al., 2020)

Line 113: Please provide the link (way) to access the Enverus DrillingInfo.

Line 178: "XCH4" should be "$XCH_4$".

Line 389: Does the "R" represent the correlation coefficient?

---

## Author Response (AR2)

**Response to referee comments on "Satellite quantification of oil and natural gas methane emissions in the US and Canada including contributions from individual basins"**

We thank the two referees for their careful reading of the manuscript and the valuable comments. This document is organized as follows: the Referee's comments are in *italic*, our responses are in plain text, and all the revisions in the manuscript are shown in blue. **Blue text** here denotes text written in direct response to the Referee's comments. The line numbers in this document refer to the updated **WORD** manuscript.

**Reviewer 1**
*Line 87: "−3.4 ± 5.6" and "US (Lorente et al., 2020)*

**Response**. We have updated the reference. Thanks.

*Line 113: Please provide the link (way) to access the Enverus DrillingInfo. Line 178: "XCH4" should be "XCH4".*

**Response**. Corrected, thanks.

*Line 389: Does the "R" represent the correlation coefficient?*

**Response**. We have clarified it in text, thanks.

**Reviewer 2**

*The line numbers refers to the tracked version of the manuscript.*

*Line 375: It was referred to as "line 385" in the responses. The newly added statement "The posterior corrections from using wintertime data are slightly different in Canada and Northeastern US because of the low observation density and low averaging kernel sensitivities" seems cursory and not exactly accurate. The western Canadian basin moves downwards in DJF and SON, upwards in MAM; the Marcellus moves downwards in DJF and MAM but upwards in JJA and SON (quite significantly with large AK sensitivity). Do those indicate anything about the seasonality of assumed emissions? I assume only an annual mean emission is inverted. Please clarify.*
**Response**. Now we clarify this in the text.
Line 241. We also calculated posterior emissions from the O/G sector using TROPOMI observations in different seasons. Here the prior O/G emissions remain constant over different seasons, so any changes in the posterior correction factors are determined by the inversion of the satellite data.

*Please provide the Varon et al. 2022 reference.*
**Response**. Added, thanks.

*Figure 4: R is shown in the figure but the caption still says R2.*
**Response**. Corrected, thanks.